# Analysis of Optimal Shift Pattern Based on Continuously Variable Transmission of Electric Vehicle for Improving Driving Distance

**Wootaek Kim** [1], **Daekuk Kim** [2], **Dokyeong Lee** [2], **Iljoo Moon** [2] **and Jinwook Lee** [3,*]

1  Department of Mechanical Engineering, Graduate School, Soongsil University, Seoul 06978, Republic of Korea
2  Department of Mechanical Engineering, Undergraduate School, Soongsil University,
   Seoul 06978, Republic of Korea
3  School of Mechanical Engineering, Soongsil University, Seoul 06978, Republic of Korea
*  Correspondence: immanuel@ssu.ac.kr; Tel.: +82-2-820-0929

**Abstract:** Extending the maximum driving distance of an electric vehicle would be beneficial given these vehicles' long charging time and limited battery capacity and to improve electricity consumption. Research on the optimal development of the powertrain is essential because the efficiency of an electric motor varies according to the operating point. In this study, we developed a simulation analysis model with a continuously variable transmission. based on the commercial electric vehicle (Company B) model and using actual vehicle driving data supplied by the Argonne National Laboratory. The shift pattern of the continuously variable transmission was then optimized by considering the change in the operating point, constant motor output area, and transmission response speed. In addition, a performance comparison was made using the model with a single reducer. The results obtained by this study showed that electronic economy improved by approximately 5% when the continuously variable transmission was applied through the combined driving simulation. Furthermore, the time taken to accelerate 0–100 km/h and 80–120 km/h reduced by 15% and 6%, respectively. The maximum driving distance on a single charge improved by 7 km. It was confirmed that the driving performance of an EV with continuously variable transmission could be improved by downsizing the electric motor to reduce manufacturing costs.

**Keywords:** electric vehicle (EV); continuously variable transmission (CVT); transmission ratio; driving distance per charge; electric economy



## 1. Introduction

Recently, renewable energy has been considered a key solution to the problems of air pollution, global warming, and oil price instability. Transportation, as the main consumer of fuel, is facing a new paradigm in electricity generation. Electric vehicles (EVs) are equipped with a rechargeable secondary battery and use electricity as a power source. They do not emit exhaust gas while being driven and produce less noise and vibration than an internal combustion engine (ICE). Compared with an ICE, the EV has excellent efficiency in urban areas, where processes, such as idle acceleration, change rapidly owing to the narrow high-efficiency section in the power operating range. EVs in urban traffic, urban-only EVs, benefit greatly from regenerative braking.

The disadvantages of EVs include their high price and the short distance travelled on a single charge. For long-distance travel, the efficiency of EVs decreases as the benefit of braking is less and the need for more battery energy storage increases [1]. To deal with these problems, most countries have established an EV subsidy system. Since 2014, this policy system has reduced the cost of EVs through different subsidy calculations based on battery efficiency, and this has encouraged the development of more competitive EVs.

In particular, EV efficiency is expected to be prioritized to reduce carbon emissions through life cycle assessments. Cheong [2] conducted a comparison of performance, cost, and emissions of EVs and ICEs in long-distance driving. As a result of the study, it was confirmed that the greenhouse gas emissions of vehicle manufacturing are similar for EVs and ICEs, but ICE emissions are about 4.5 times higher than EVs when driving. In addition, when considering maintenance and energy costs, EVs become cheaper than ICEs when used for 10.5 years or more. However, it is expected that the cost reduction of EVs will rise significantly along with the technological development of batteries and electrification.

Therefore, it is necessary to develop and apply various technologies to increase the efficiency of EVs [3,4]. Compared to EVs, ICEs have a narrow engine operating range, complex transmissions and couplings with various gear ratios. Therefore, for this reason, power loss occurs due to the gear ratio in the energy transfer step. In contrast, since the motors in EVs have wide operating range and high initial torque, it is possible to drive them with only a simple reducer (transmission). Due to the high initial torque, transients can be overcome quickly. Currently, most EVs have a fixed single ratio because they operate more widely than ICEs and can utilize the maximum torque at lower rotational speeds. Research on the applications of EV transmission is being actively conducted for various purposes. The transmission in a vehicle gradually expands the driving range of the driving source and moves the operating point of the driving source to the high-efficiency section, especially at low speeds, to increase the torque and maximum speed. In the past, the transmission was an essential component of vehicles with an ICE. However, in the case of EVs, the operation range is wider than that of an ICE, and the efficiency is generally high in the entire range; hence, only a single reducer is applied. However, owing to the recent expansion of the EV market, manufacturers, such as Porsche (Taikan) and Bosch (CVT4EV), have increased their research on EV transmission to remain competitive.

Automotive transmissions are divided into manual and automatic transmissions. These include clutch-type manual transmission (MT), torque converter automatic transmission, automatic manual transmission, dual-clutch transmission, and continuously variable transmission. Unlike other transmissions, the CVT does not require multiple gears to implement each shift stage; hence, it has a small volume and does not lose power with changes in the gear ratio. These factors ensure smooth acceleration, which is an advantage of EVs.

CVTs are divided into push-type CVTs that use belts, pull-type CVTs that use chains, and friction-type, tidal type, and E-CVTs that use planetary gears. According to the operating principle of the belt-type CVT, the belt connects the driving pulley and passive pulley, and each pulley is a basic component. Cone-shaped disks are arranged facing each other to change the rotation diameter of the belt between the pulley through tension–compression to realize the shift ratio. Although the belt-type CVT has the advantage of reduced noise, power loss occurs owing to slip between the pulley and the belt. On the other hand, the chain-type CVT has increased power transmission efficiency because slip is reduced by replacing the belt with a chain, and it also responds to a relatively high torque. Its disadvantage is the noise generated by the friction between the pulley and the chain.

In the study by Park [5], a model of a synchronizer-type two-speed transmission was designed using MATLAB/Simulink. The acceleration performance, maximum speed, and efficiency were determined to obtain shift pattern and improve electronic fuel economy by up to 4%, compared with a Nissan Leaf.

Ren et al. [6] studied the electronic fuel economy of each driving mode by applying a CVT's 1st, 2nd, 3rd, and 4th transmissions to 40 kW motorized EV analysis models. For the 4th transmission model, 4.5% FTP-75 to 8.0% CVT and 5.3% FTP-75 to 10.9% improvements in the New European Driving Cycle were obtained.

Kwon et al. [7] developed shift control, state of charge (SOC), drive resistance calculation, and vehicle dynamics models using MATLAB/Simulink to select shift pattern at the point with lowest SOC consumption after driving in test mode.

Kim et al. [8] developed an EV analysis model using AVL CRUISE software and a two-speed transmission. The comparison of efficiency between a single-reducer electric model with an 80 kW motor and a second-speed transmission EV with a 60 kW motor confirmed the possibility of reducing motor capacity through transmission.

Shin and Bang [9] developed a regenerative braking system for an EV that combined a conical cone with a roller and flywheel. A prototype was manufactured, which confirmed increase in energy storage and regenerative braking efficiency.

Hofman et al. [10] developed an analysis model for Volkswagen's Lupo 3 L and compared the application of an MT and CVT. They confirmed that the CVT provided energy savings of 7% compared with the MT when the final driving ratio was reduced.

Ahsan et al. [11] applied a multistage transmission to an EV. They confirmed an up to 90% improvement in maximum speed, 19.4% acceleration performance, and 11.3% energy savings when a four-speed transmission system was applied to an electric bus.

Francesca et al. [12] developed vehicle performance and energy efficiency calculation functions using MATLAB/Simulink. They confirmed that the multistage transmission application model could reduce energy consumption by 29% compared with the single transmission application model.

Considering the above previous studies, it is important to improve the efficiency of an electric motor by moving the operating point to reflect the actual motor efficiency map in an analysis model. That is, the operating point of the motor changes according to the change in vehicle speed. If the operating point of the motor changes, it is highly likely that the efficiency of the motor does not operate at the optimal point. Therefore, if the CVT is used, we can set it to operate in the section where the operating efficiency of the motor is maximized. Of course, all transmissions generate a delay according to operation, and how quickly they respond to it is a factor that has a major impact on fuel economy. When the motor passes through a certain torque range and proceeds at high RPM, a certain output range proceeds, which shows a tendency for efficiency to decrease as the motor goes to high RPM due to the characteristics of the motor. Therefore, in order to maximize the efficiency of the motor, to prevent the RPM from rising above a certain output range, and to compensate for this, the maximum operating RPM of the motor is limited to a specific RPM using the transmission ratio of the CVT.

However, previous studies have been insufficient in this regard, which has resulted in large differences between the analysis model data and the actual vehicle data and CVT shift range. In other words, the approximate electronic fuel efficiency increase or decrease can be identified but the reliability is limited.

In this study, a simulation analysis model was developed using a CVT based on a commercial EV (Company B) model and actual driving data from the Argonne National Laboratory (ANL). ANL is the largest renewable clean energy/nuclear physics research institute in the United States. Then, the shift pattern of the CVT was optimized by considering the operating point change, constant output area, and transmission response speed. In addition, a performance comparison was made by applying a single reducer to the analytic model.

## 2. Research Methods

### 2.1. Research Procedure

Figure 1 shows the full process of this study. First, an EV analysis model was developed using the CRUISE software (AVL, Graz, Austria), based on actual difference data on Company B's EV to ensure reliability. The CRUISE program is a vehicle driving simulation software that achieves the best balance of efficiency, exhaust gas, and vehicle performance when driving. It also has the advantage of a high calculation speed. The actual efficiency map of the drive motor was derived and reflected in the analysis model using the FluxMotor software (Altair, Troy, MI, USA), which has a simple layout and fast operation. Its reliability was verified by comparison with actual vehicle driving data from the ANL. Based on this, a CVT-applied EV analysis model was developed. Then, a shift pattern design

strategy was established to design the optimized shift pattern for improved efficiency. The electronic fuel efficiency, according to the transmission response speed, electronic fuel economy, acceleration performance, and climbing performance of the CVT-applied EV were compared and analyzed. In this study, the 1st/top 6 was a common value. In some models, the 1st/top expanded to 7.5 by combining two-speed transmissions. However, because the lowest and highest ends could not be changed continuously, the 1st/top 6 was selected in this study. The single cycle test (SCT) method was applied to calculate the EV distance. The SCT method calculates the composite electronic fuel economy of driving in the FTP-75 and HWFET modes to achieve an SOC of 0% on the chassis dynamometer. This refers to the relational equation applied to urban driving and highway driving energy consumption efficiencies verified with the FTP-75, C-FTP75, HWFET, SC-03, and US-06 (Five-cycle), similar to the overall electronic fuel economy measured in the FTP-75 and HWFET modes. In the case of an EV, the correction coefficient of 0.7 in the FTP-75 and HWFET modes should be multiplied to reflect the actual driving conditions. The reason for this is because the distance per charge could significantly vary depending on the temperature of the environment, battery status, and driving conditions.

---

### Development of CVT-applied EV analysis model (AVL Cruise)

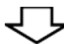

### Optimization of shift pattern

**- Shift pattern design considering motor operating point change**
**- Shift pattern design considering the constant output area**

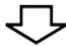

### Performance comparison

**- Comparison of electronic fuel efficiency according to CVT response time**
**- Comparison of power/performance with single reducer analysis model**

**Figure 1.** Research process of this study.

The combined electronic fuel economy based on the five-cycle correction formula was calculated with the following contents:

Composite maximum single-charge distance (km) = 0.55 × city driving single-charge distance + 0.45 × highway driving single-charge distance     (1)

Charging distance on urban driving = Single-charge distance obtained while continuously driving in accordance with the Urban Dynamometer Driving Test Plan (UDDS) in the 0.7 × FTP-75 mode     (2)

Charging distance on the highway = Single-charge distance obtained by continuously driving in the 0.7 × HWFET mode     (3)

Combined electronic fuel economy (km/kWh) = 1/(A + B)     (4)

A = 0.55/Electronic fuel consumption rate according to repeated UDDS driving in FTP-75

B = 0.45/Electronic fuel consumption rate according to repeated HWFET driving

## 2.2. Analytic Modeling

Basic Model for EV

To ensure the reliability of the research results, an analysis model was developed that reflects the actual vehicle specifications to enable the comparison between the analysis model and actual data. To determine the efficiency improvement by changing the operating point, an efficiency map was derived using the FluxMotor (Altair) software based on the specifications of the driving motor installed in Company B's vehicle. An analysis model was developed based on the efficiency map and specifications.

Figure 2 shows the layout of the EV analysis model in the AVL CRUISE software based on the specifications of EV listed in Table 1. Table 2 shows the EV model parameters used in the AVL CRUISE program.

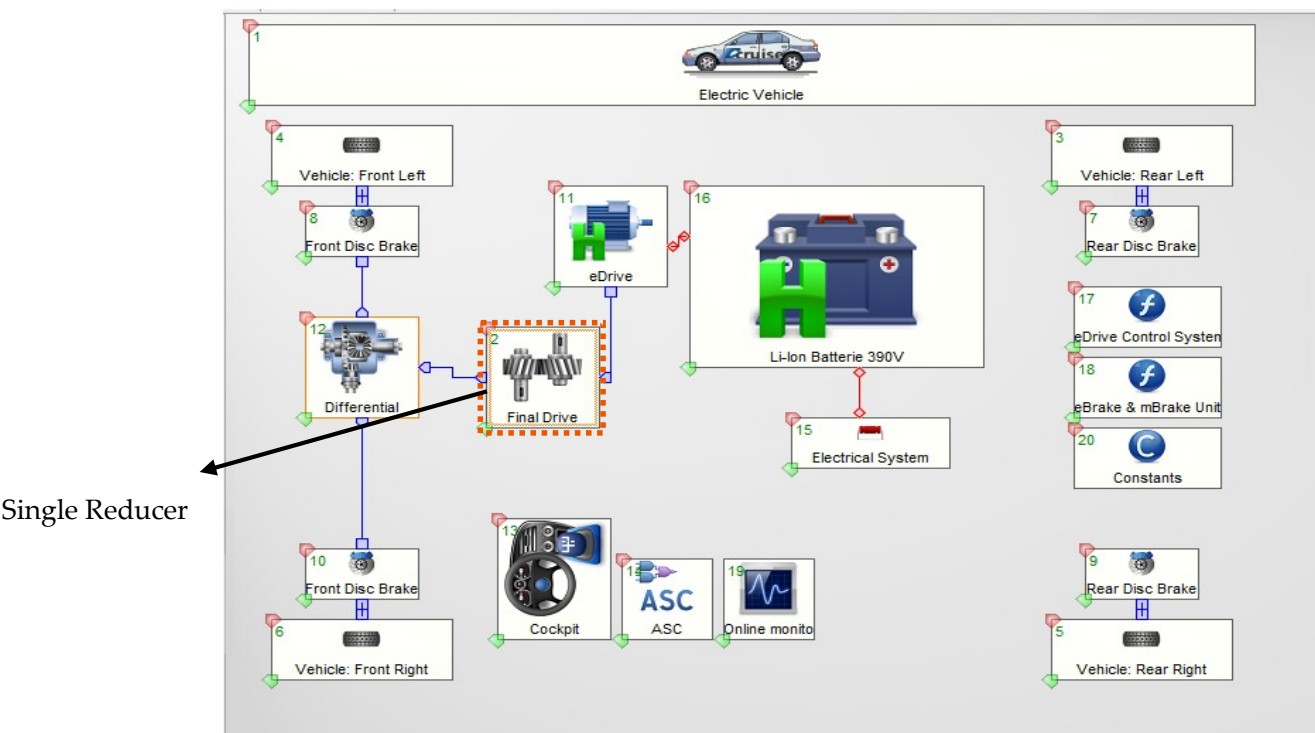

**Figure 2.** EV model layout developed by this study.

**Table 1.** Specifications of EV used in this study.

| Item | Unit | Data |
|---|---|---|
| Vehicle length | m | 4008 |
| Vehicle width | m | 1775 |
| Vehicle height | m | 1578 |
| Frontal area | m$^2$ | 2.38 |
| Wheelbase | m | 2570 |
| Curb weight | kg | 1297 |
| Single gear ratio | - | 9.7 |
| Drag Coefficient | - | 0.29 |
| Tires | - | 155/70 R19155/70 |
| Battery capacity | kWh | 22 (60 Ah) |

**Table 2.** EV model parameters used in AVL CRUISE program.

| Main Factor | Input Variable | Data |
| --- | --- | --- |
| Electric Vehicle | Drag Coefficient | 0.36 |
| | Frontal Area | 2.3 m$^2$ |
| | Curb Weight | 1340 kg |
| | Gross Weight | 1550 kg |
| | Wheel Base | 2570 mm |
| E-drive | Type of Machine | PMSM |
| | Normal Voltage | 390 V |
| | Maximum Speed | 11,400 (1/min) |
| Li-Ion Batterie 390V | Maximum Charge | 15 Ah |
| | Initial Charge | 91.8% |
| | Maximum Voltage | 400 V |
| | Minimum Voltage | 220 V |
| | Number of Cell-Rows | 5 |
| | Ohm Resistance | 0.8 Ω |
| eDrive Control System | Operating motor, | - |
| | Brake control unit | - |
| Final Drive | Transmission Ratio | 9.7 |

Based on Company B's EV drive motor specifications in Table 3, the electric motor modeling and performance analysis software (Altair FluxMotor) was used to reflect the motor efficiency maps and torque–output curves in the CRUISE software (AVL) analysis models. Figure 3 shows the efficiency contour map of the motor reflected in the analysis model. Figure 4 compares the maximum torque–power curve derived through analysis using actual data on Company B's EV drive motor. Figure 5 shows the degree of voltage drop depending on the battery discharge. The experimental and calculated motor performance showed good agreement.

**Table 3.** Electric motor parameters used in Altair FluxMotor program.

| Item | Data | Item | Data |
| --- | --- | --- | --- |
| Assembly mass | 42 kg | Number of Slots | 72 |
| Slot depth | 22.0 mm | Number of pole pairs | 6 |
| Stator outer diameter | 242.1 mm | Stator turns per coil | 9 |
| Stator inner diameter | 180.0 mm | Wire size | 21 |
| Rotor outer diameter | 178.6 mm | Machine type | PM (HSM) |
| Rotor inner diameter | 60 mm | Maximum torque | 250 Nm |
| Stack length | 132.3 mm | Maximum speed | 11,400 rpm |
| Tooth width | 5.0 mm | Voltage range | 250–400 V |
| Slot opening | 1.7 mm | Max phase current | 400 A |

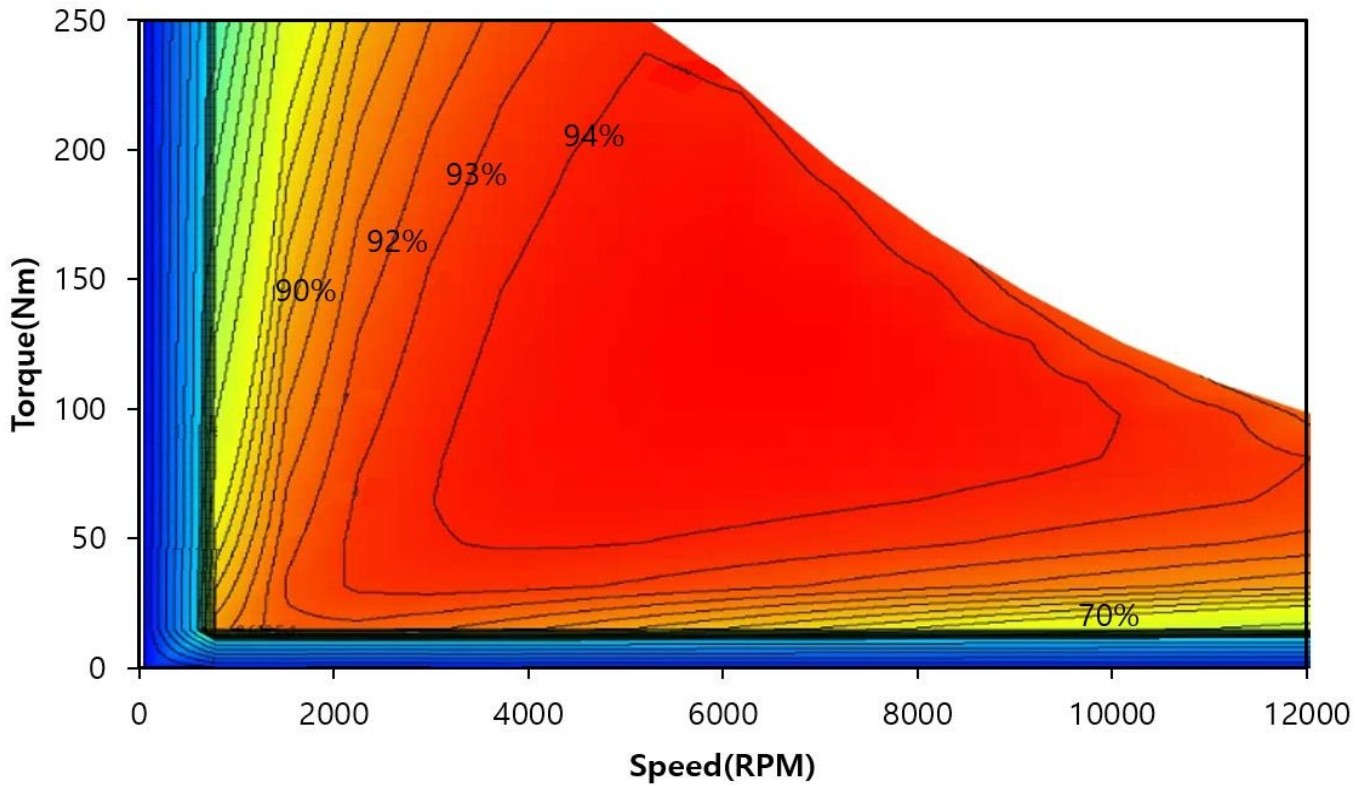

**Figure 3.** Efficiency contour map of electric motor used in this study.

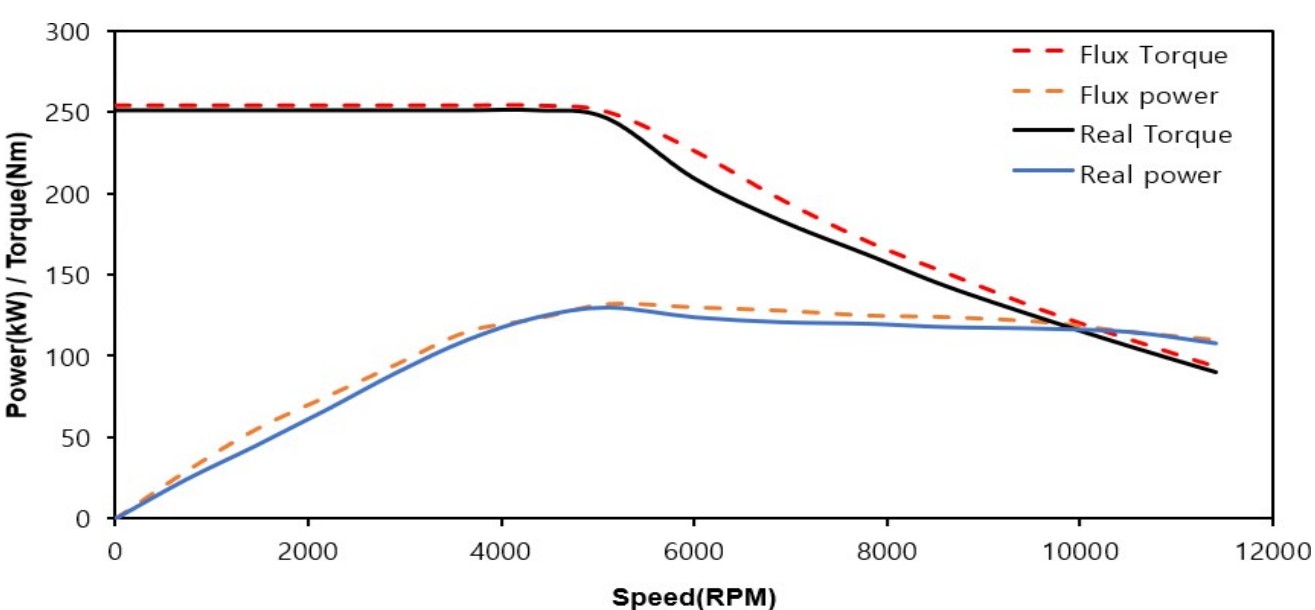

**Figure 4.** Comparison of motor torque–output.

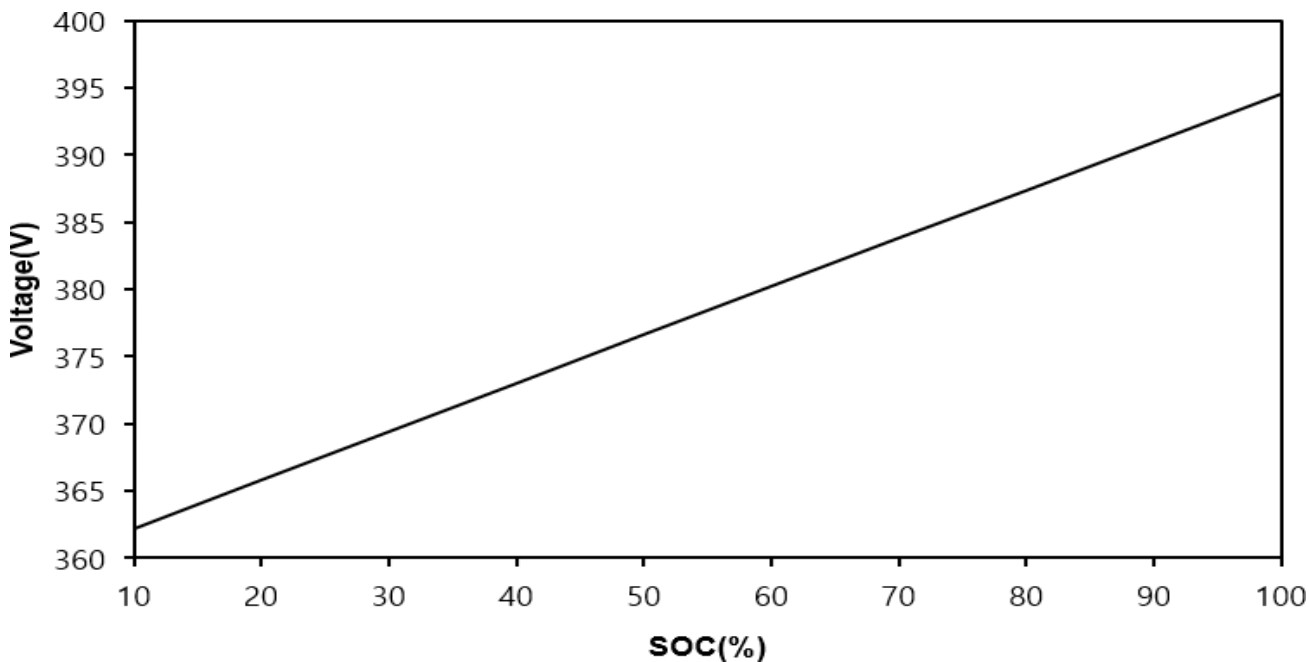

**Figure 5.** Battery voltage–SOC drop curve.

*2.3. Analysis of EV Model with CVT*

CVT-Applied EV Modeling

Figure 6 shows the CVT-applied EV analysis model. The model was developed by installing a CVT and CVT control in the EV analysis model, which was manufactured based on Company B's EV. Since the CVT typically has a shift range of 0.5:1 to 3:1, the CVT shift module was implemented by applying a single reducer to the rear of the CVT to enable its use within the appropriate range for vehicle operation.

The CVT was characterized such that, as the transmission ratio increased. according to the operating method, additional power was required to compress the pulley, resulting in power loss. Therefore, although there was a difference in the transmission efficiency of the CVT, depending on the driving speed and torque, only the efficiency according to the transmission ratio was reflected, as shown in Figure 7. The CVT efficiency curves of related studies were referred to in consideration of the user environment of the analysis model, and the CVT efficiency applied was applied to the Cruise software analysis model.

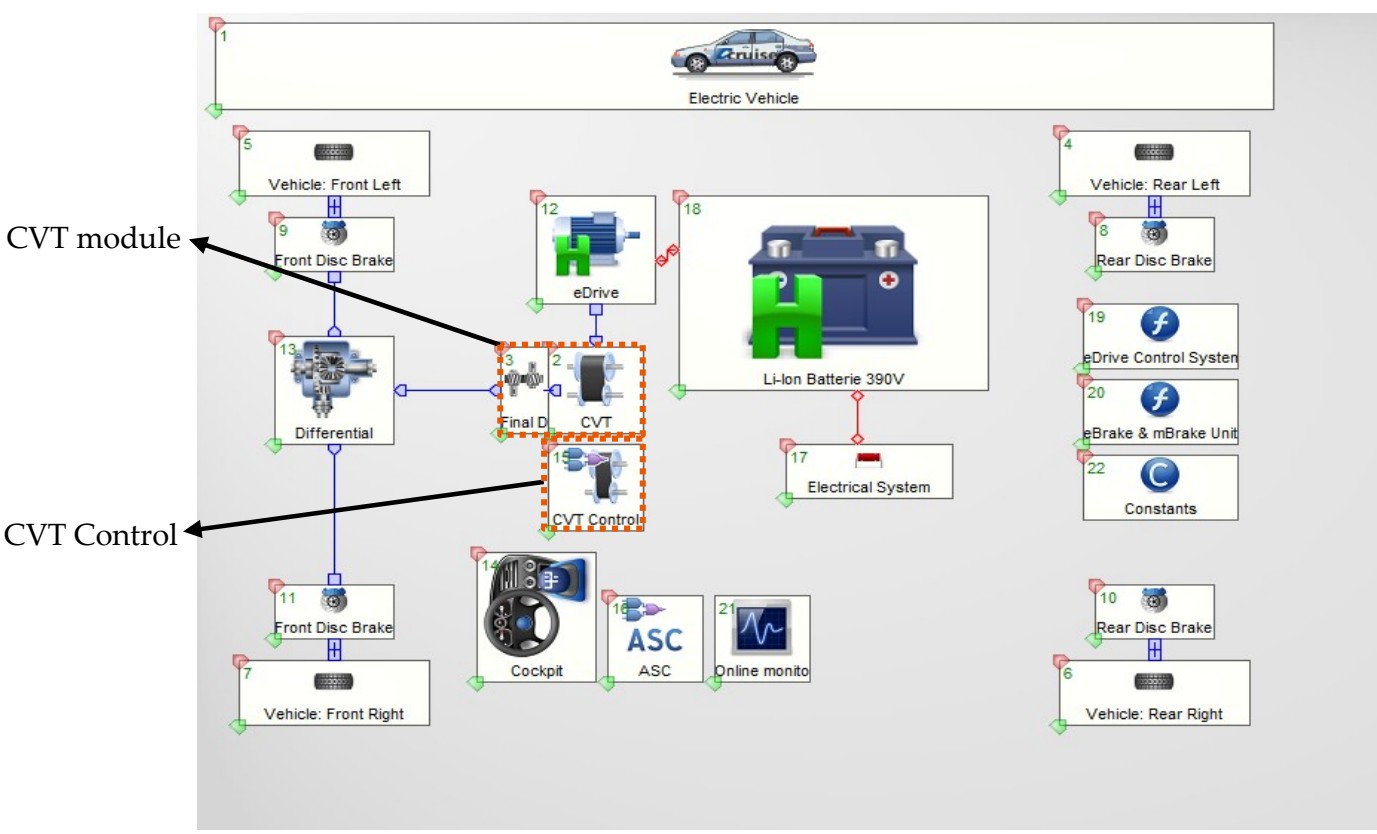

**Figure 6.** CVT-applied EV model layout developed by this study.

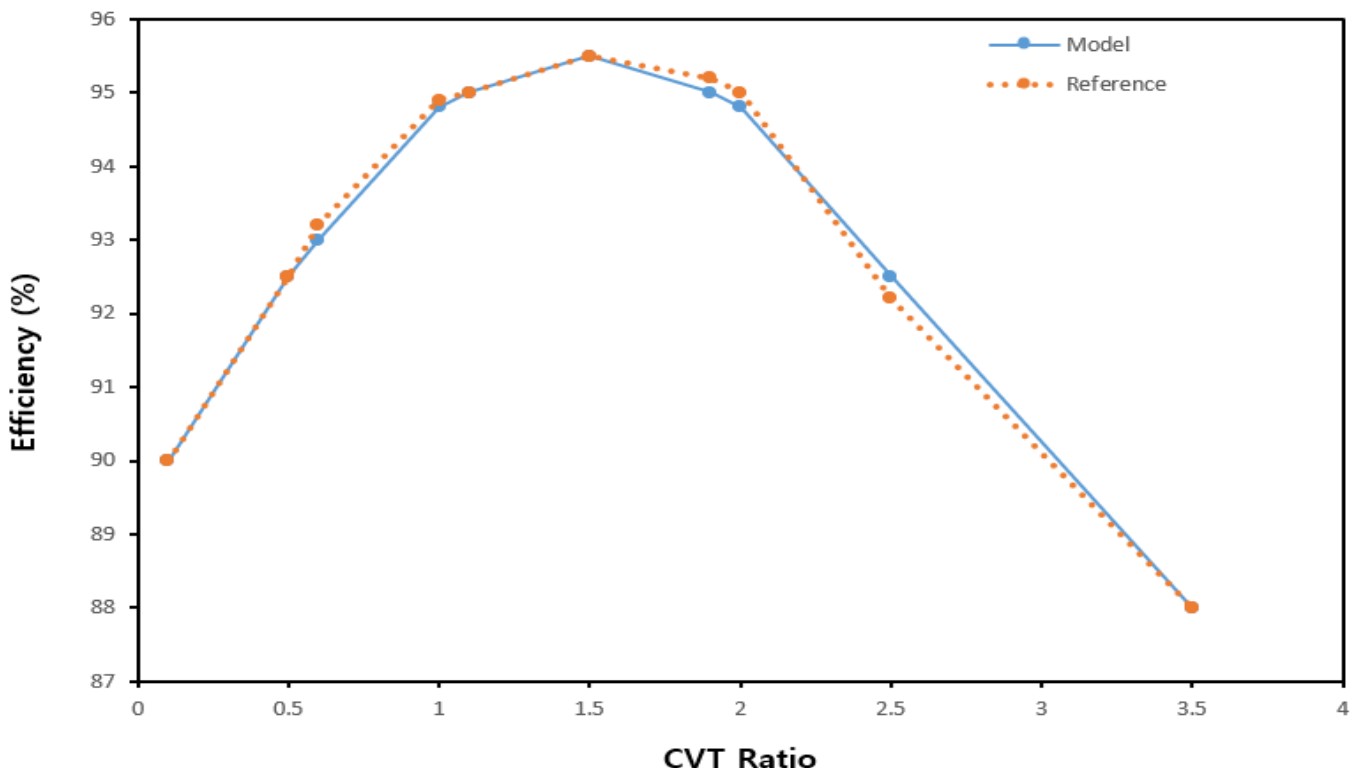

**Figure 7.** Efficiency of CVT according to shift ratio.

## 3. Results and Discussion

### 3.1. Comparison with ANL Experimental Data

To verify the reliability of the analytical model, we compared the actual vehicle data on Company B's EV, provided by the ANL, with the SOC reduction and voltage–current increase and decrease in the test-driving mode.

As the initial SOC in the actual experiment was 91.8%, the initial SOC in the analysis model was also set as 91.8%. Figure 8 shows that the changes in the SOC, voltage, and current while driving were consistent with the actual vehicle data on the amount of current; therefore, the reliability of the model was confirmed. Table 4 lists the performance values obtained from the analysis model. The relevant experiment and analysis assumed a room temperature condition; hence, there was a slight numerical difference in the authorized electronic fuel economy combining low- and high-temperature electricity.

**Table 4.** EV performance obtained by analysis.

| Item | Unit | Data |
|---|---|---|
| Electronic economy | km/kWh | 5.474 |
| Driving Distance per charge | km | 139.07 |
| 0–100 km/h arrival time | s | 6.98 |

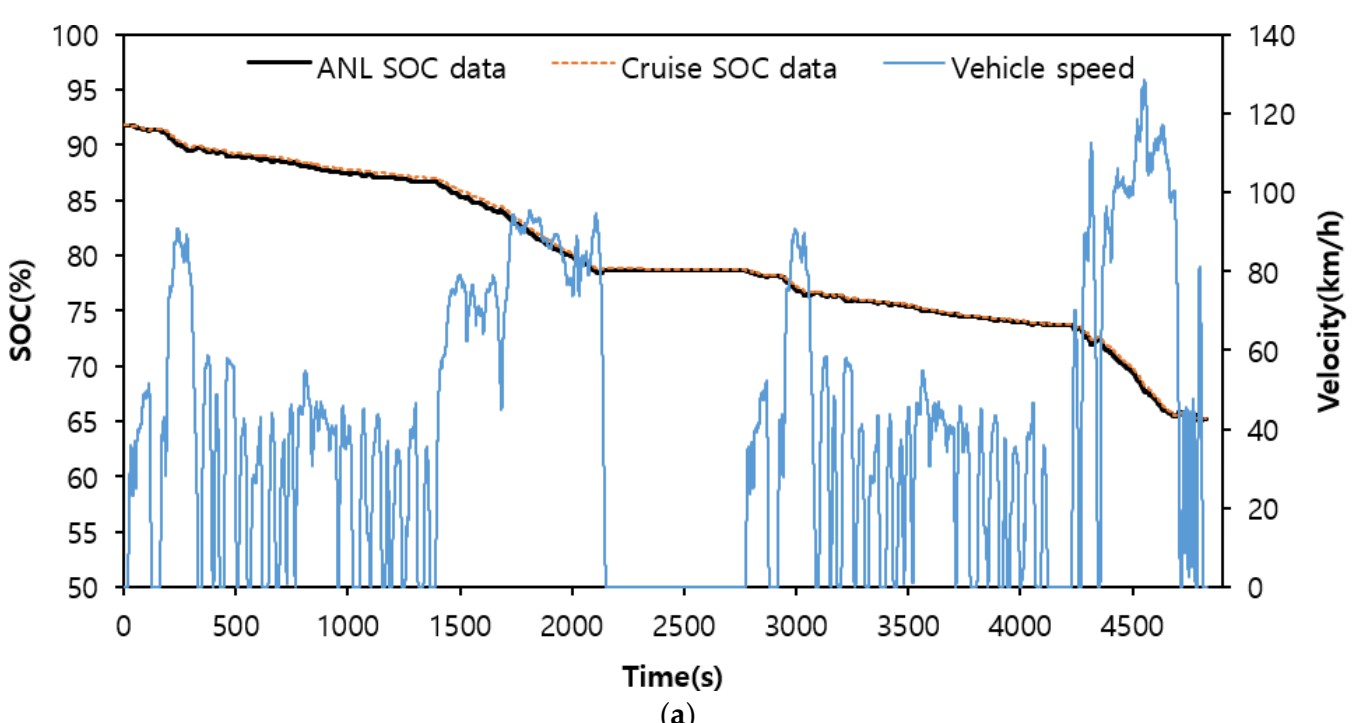

(a)

**Figure 8.** *Cont.*

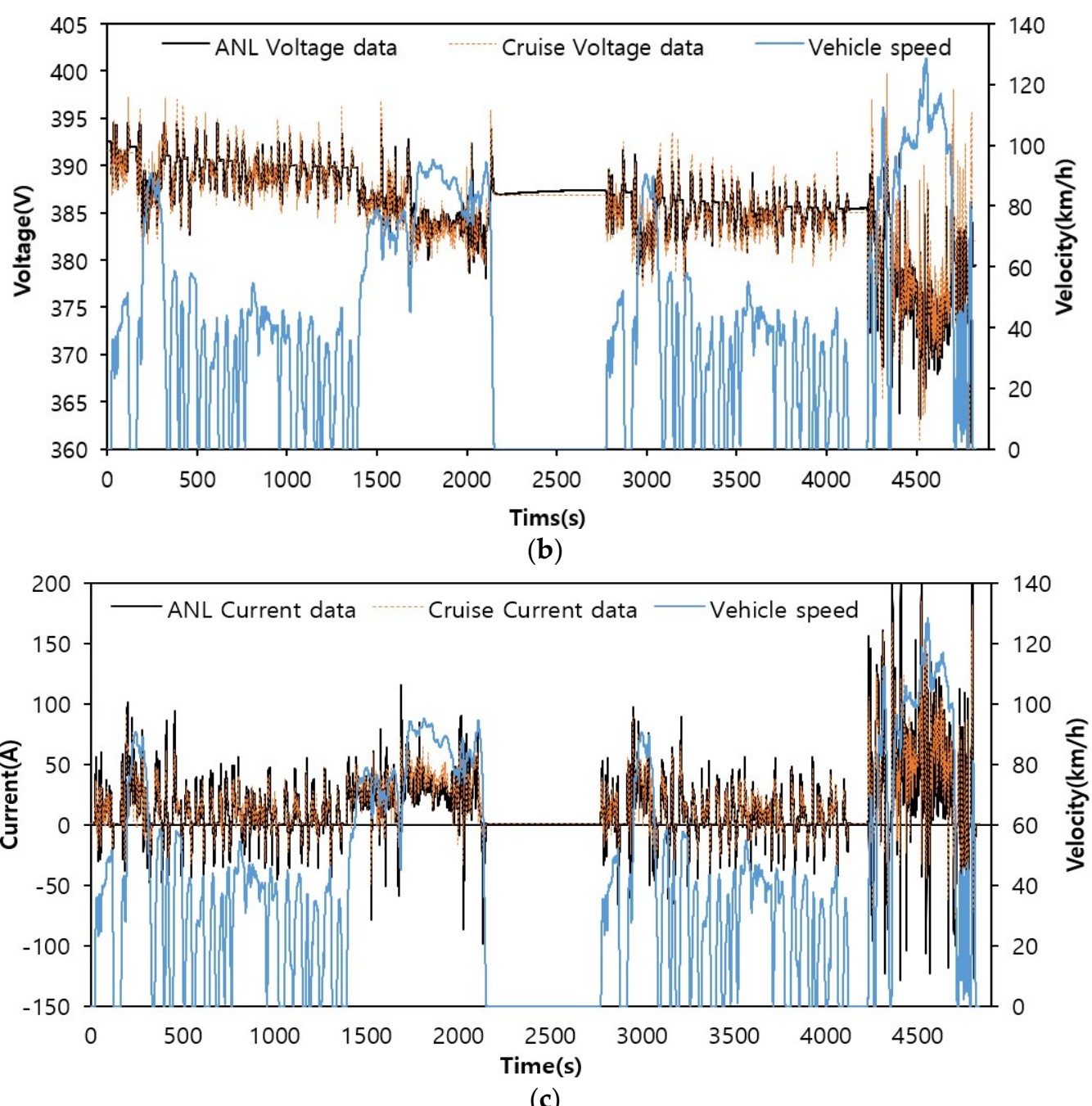

**Figure 8.** Comparison with ANL experimental data. (**a**) SOC, (**b**) voltage, and (**c**) current.

*3.2. CVT-Applied Analysis Model Result*

Based on the SCT method proposed by the Ministry of Environment of Korea, this study measured the ratio of the FTP-75 cycle in the city driving mode and HWFET cycle in the highway driving mode. The combined electronic fuel economy was calculated by reflecting 55% and 45% of the EV electronic fuel economy measurement correction coefficient, 0.7 to derive the final composite electronic fuel economy and total mileage. Figures 9 and 10 show the speed profiles of each mode.

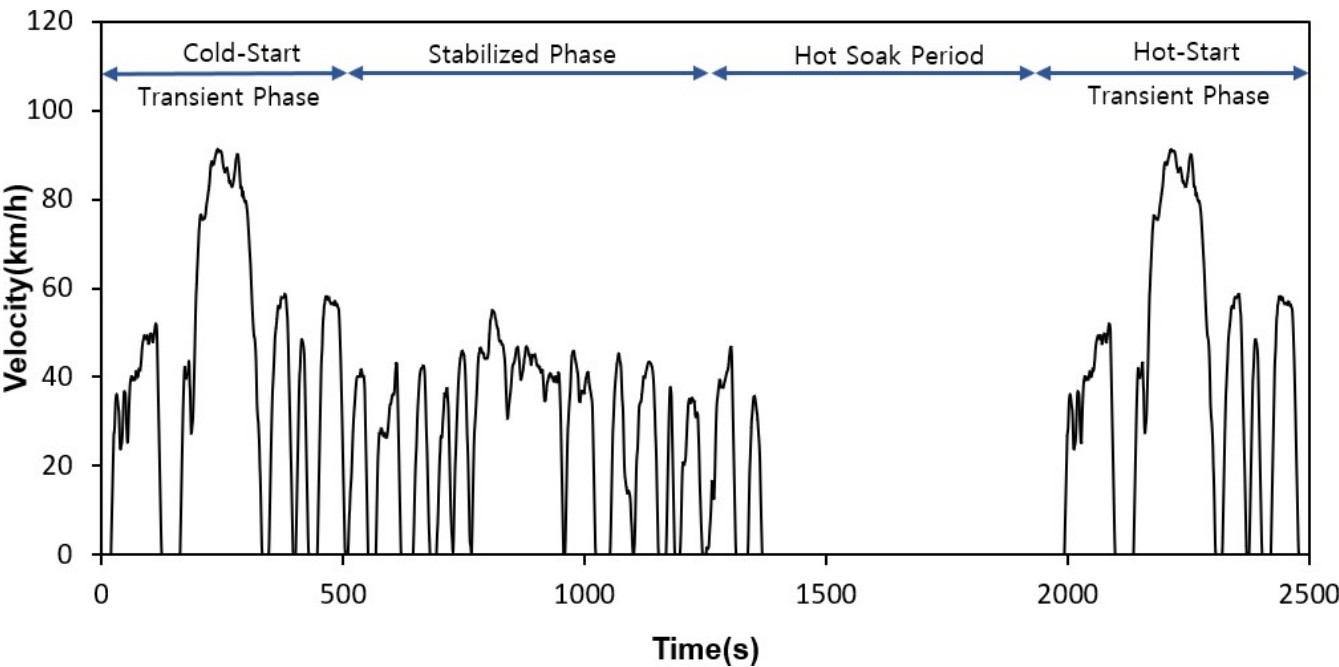

**Figure 9.** FTP-75 driving cycle.

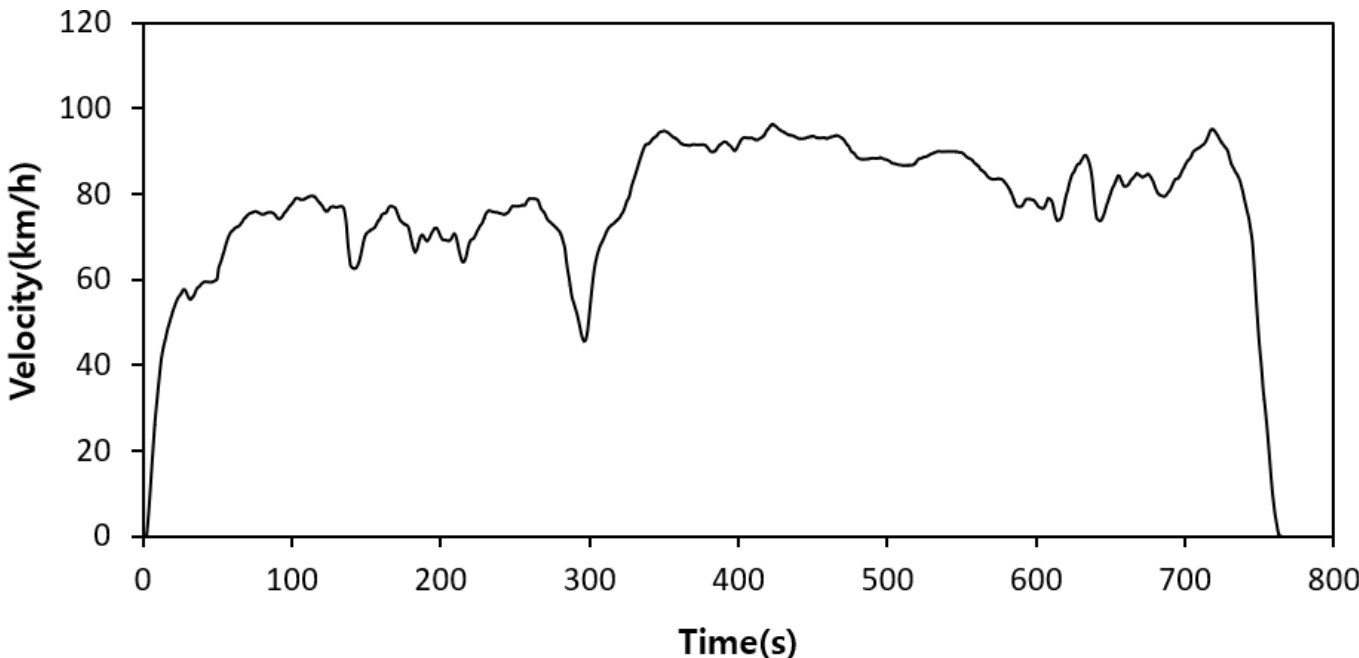

**Figure 10.** HWFET driving cycle.

3.2.1. Optimization of Shift Pattern by Considering the Motor Operating Point Change

Figure 11 shows how to change the operating point. The analysis of the operating point of the EV analysis model with a 9.7:1 single reducer showed that, to increase the torque, the operating point had to be changed by reducing the operating RPM of the motor owing to the low driving torque at 2200 rpm or higher.

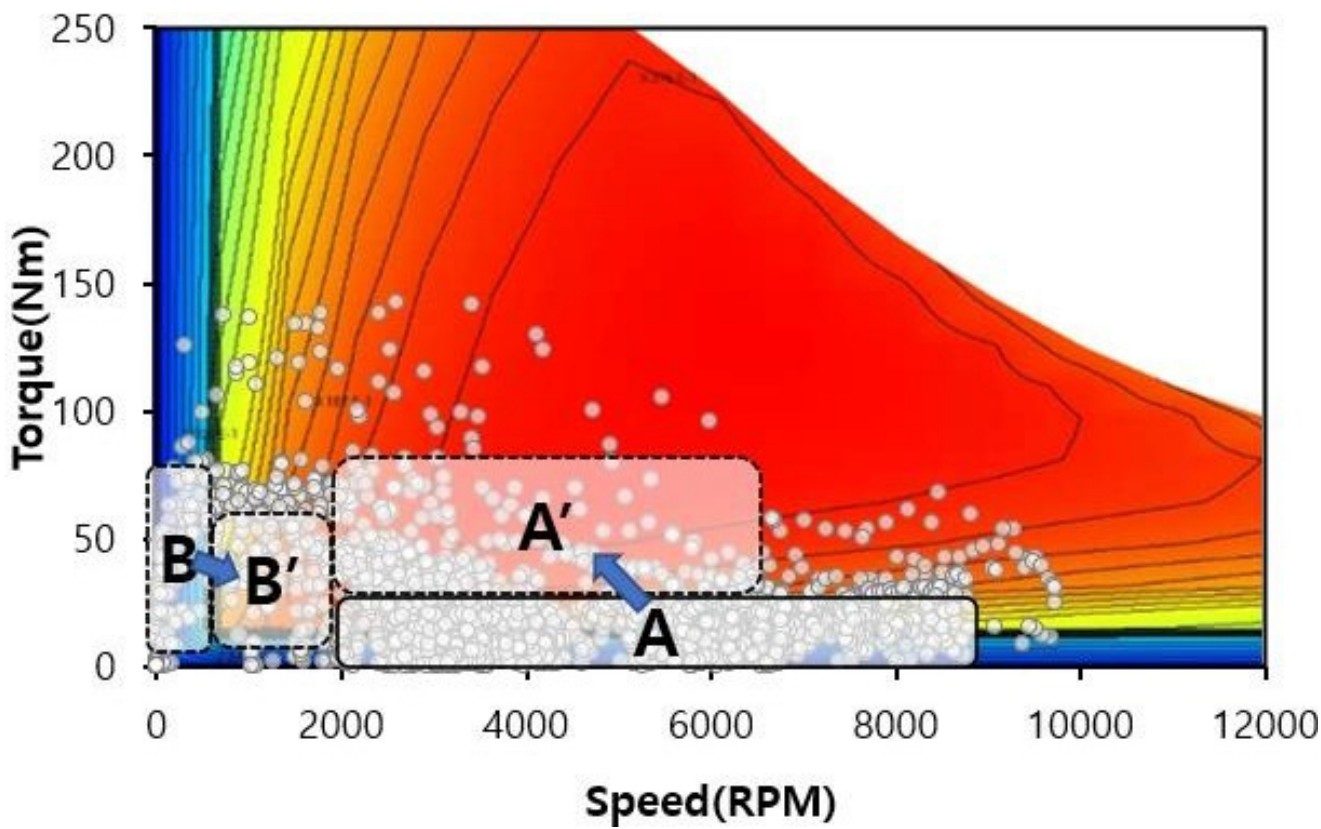

**Figure 11.** Strategy of optimal shift pattern in EV with CVT for improving the driving distance.

Table 5 lists the shift range according to the deceleration factor variable for the [A–A']
movement and the deceleration ratio of the single reducer for its implementation. Based on
the deceleration ratio (9.7:1), the lowest shift ratios of the CVT were 1/1, 5, 1/2, 5, 1/3, and
1/4. The highest shift ratio of the CVT was derived by multiplying the CVT span by 6.

**Table 5.** Comparison of CVT shift range and single-reducer ratio according to deceleration ratio.

| | Reduction Ratio | | | | | |
|---|---|---|---|---|---|---|
| | $n = 1.5$ | $n = 2$ | $n = 2.5$ | $n = 3$ | $n = 3.5$ | $n = 4$ |
| Min. ratio | 6.5:1 | 4.85:1 | 3.85:1 | 3.25:1 | 2.75:1 | 2.45:1 |
| Max. ratio | 39:1 | 29.1:1 | 23.1:1 | 19.5:1 | 16.5:1 | 14.7:1 |
| Single reducer ratio | 13:1 | 9.7:1 | 7.7:1 | 6.5:1 | 5.5:1 | 4.9:1 |

At 2200 rpm or less, the operating point must be changed to reduce the torque by
increasing the RPM through a high deceleration ratio. Therefore, the minimum motor
rotational speed was limited to 1600 rpm, based on the efficiency map of the motor for the
[B–B'] movement. However, when $n = 3, 3.5$, and 4, because the minimum motor rotational
speed at 10 km/h was 1600 rpm or less, the minimum motor rotational speed at 10 km/h
was restricted to 1200 rpm.

Figure 12 shows the composite electronic fuel economy values measured by applying
a single deceleration ratio, according to the deceleration magnification based on Table 5.
The FTP-75 and HWFET modes were measured by the SCT method, similar to the method
for calculating the electronic fuel economy and maximum driving distance of the analysis
model. The correction factor was multiplied by 0.7. According to the electronic fuel
economy measurements, the maximum value was $n = 3.5$. Therefore, this study was
conducted based on the value in which the magnification of the single reducer was set to
5.5.

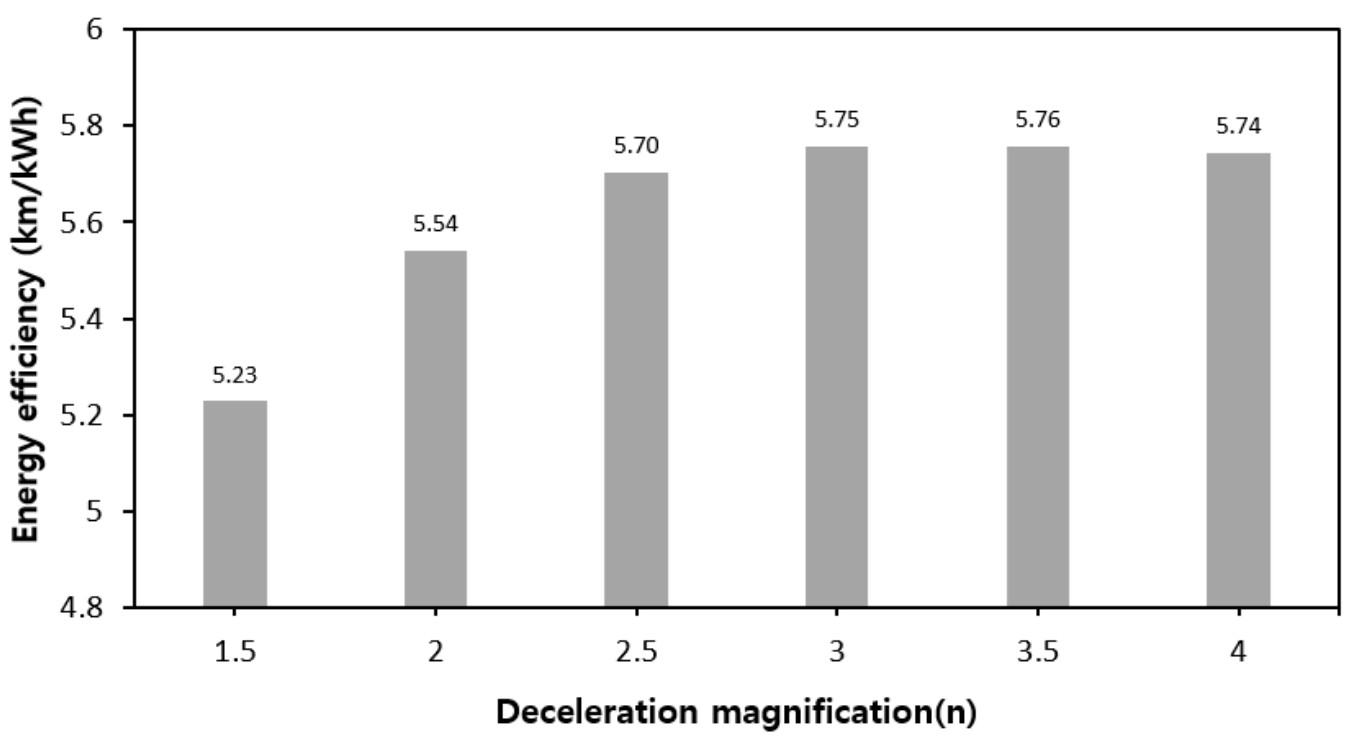

**Figure 12.** Comparison of energy efficiency according to deceleration ratio.

In summary, a CVT-applied model was developed by applying a CVT, based on the EV analytic model, to verify the reliability. The operating point of the electric motor was expected to change due to the combination of the CVT and single-reducer shift ratio; hence, the RPM was increased using a high shift ratio in the high-torque low-RPM section, and the torque was increased using a low shift ratio in the low-torque high-RPM section. Starting from 1/1.5, the existing deceleration ratio of 9.7 was linearly divided by the 1/4 ratio, and the maximum shift ratio was derived by multiplying the CVT span by 6. The change in the operating point of the motor was optimized when $n = 3.5$.

### 3.2.2. Comparison of Electric Economy According to Response of CVT

One of the main performance criteria of the transmission is the time required for transmission. To maintain the shift pattern at the optimum efficiency point, the transmission should be implemented as quickly as possible, depending on the driver's needs and the vehicle's driving state. Figure 13 shows the change in the electronic fuel economy depending on the response speed of the transmission. It was confirmed that, as the response time increased, the electronic fuel economy decreased.

### 3.2.3. Result of Electric Economy

Table 6 shows the comparison of the results of the preparation for the without CVT model and the CVT-applied model. Each measurement method had an initial SOC of 91.8% while driving in the FTP-75 and HWFET modes, until an SOC of 0% was obtained when multiplying this by the correction factor of 0.7. For the analysis model, the composite electronic fuel economy was calculated by considering the FTP-75 55% and HWFET 45% in the same manner as that in the composite electronic fuel economy measurement method. In a single calculation, the rate of increase of the electronic fuel economy in the FTP-75 and HWFET modes improved by 2.02% and 9.44%, respectively. The combined electronic fuel economy improved by 5.11%. Despite no significant increase in rate in the urban driving mode, it was confirmed that the application of the CVT significantly increased the rate in the highway driving mode.

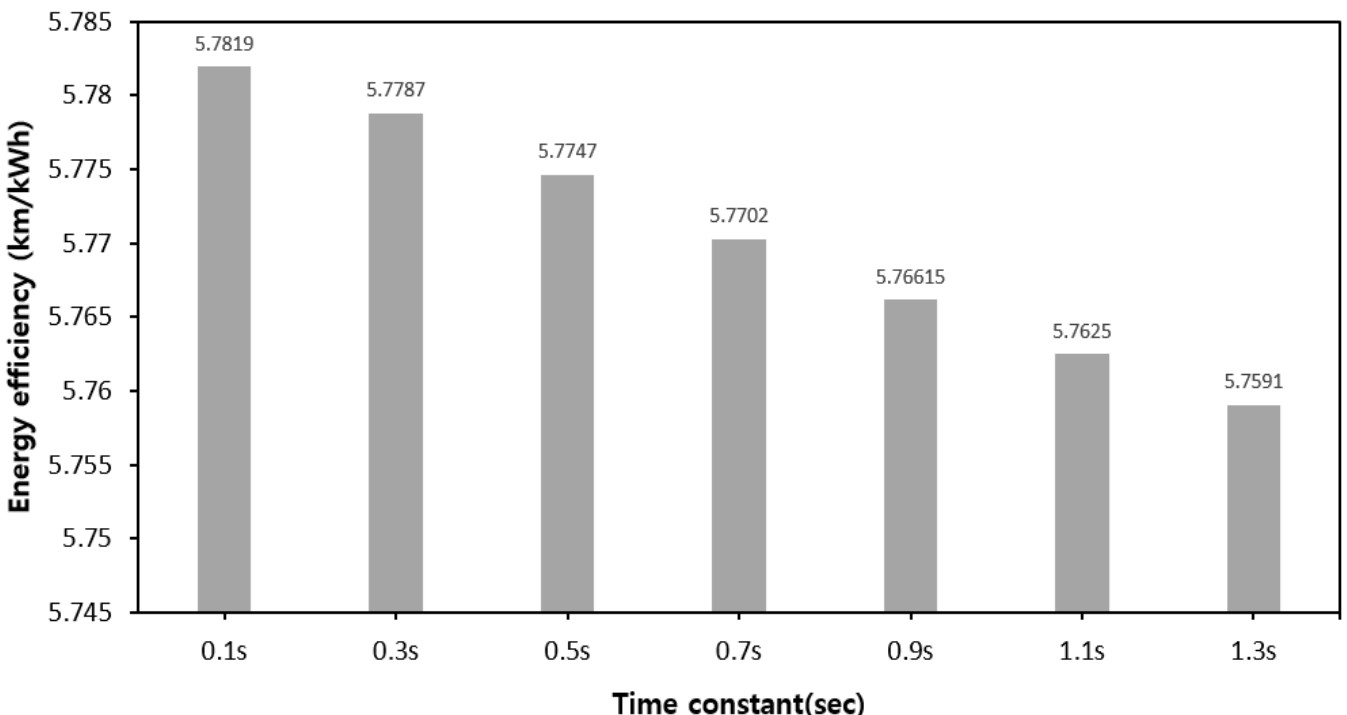

**Figure 13.** Comparison of energy efficiency according to CVT's time constant.

**Table 6.** Comparison of electric economy between driving cycles.

| | Electronic Economy (km/kWh) | | |
|---|---|---|---|
| | **Without CVT** | **With CVT** | **Improving Rate (%)** |
| FTP-75 | 5.90 | 6.02 | 2.02 |
| HWFET | 4.97 | 5.44 | 9.44 |
| Composite | 5.48 | 5.76 | 5.11 |

### 3.2.4. Result of EV Acceleration

Table 7 presents the comparison of the CVT-applied EV and single-reducer analysis model by measuring the arrival time for two speed ranges: 0–100 km/h and 80–120 km/h. When the CVT was used, the acceleration at low speed was significantly better than that of the single reducer. However, at the normal driving speed of 80–120 km/h, the degree of improvement in acceleration was not evident, as both the CVT-applied model and single-reducer model shifted in a certain output area. Therefore, it was confirmed that the CVT application significantly affected the improvements in driving performance and electronic fuel economy.

**Table 7.** Comparison of EV acceleration.

| | Acceleration Time (s) | | |
|---|---|---|---|
| | **Single Reducer** | **CVT-Applied** | **Reduction Ratio (%)** |
| 0–100 km/h | 12.56 | 10.85 | 15.76 |
| 80–120 km/h | 4.09 | 3.83 | 6.36 |

### 3.2.5. Result of Climbing Performance

Table 8 compares the acceleration at 0–100 km/h, when the slope was 10–20%, and the reduction in the arrival time. It was confirmed that the degree of reduction in the arrival time of the CVT-applied model was greater than that on flat land.

**Table 8.** Comparison of reduction degree according to speed section when slope is applied.

| | Time-Concentration (s) | | |
|---|---|---|---|
| | One Speed Reducer | CVT Applied | Reduction Ratio (%) |
| 10% | 8.58 | 7.42 | 15.52 |
| 20% | 12.48 | 10.85 | 13.06 |

## 4. Conclusions

This study examined the application of a powertrain, i.e., CVT, to improve the driving distance on a single charge of an EV. The vehicle model of Company B's EV was implemented using the CRUISE and FluxMotor software. Then, the CVT was applied to the validated model to compare the driving distance per charge, composite electric economy, and acceleration/climbing performance in the HWFET and FTP-75 driving cycle modes. The following conclusions were drawn in this study:

(1) Based on the experimental data provided by the ANL, the reliability of the EV model developed in this study was verified by comparing changes in the SOC, voltage, and current during vehicle driving. A CVT-applied model was developed by applying a CVT based on the EV analytic model to verify the reliability.

(2) In the analysis of the shift pattern, based on CVT of the electric vehicle, the operating point of the electric motor was expected to change by the combination of the CVT and single-reducer shift ratio and was optimized when $n = 3.5$.

(3) The energy efficiency of the electric vehicle with CVT improved by approximately 5.11% when continuously variable transmission was applied through the combined driving simulation. The improvement was greater in highway driving than in the urban driving modes.

(4) When the CVT was applied, the maximum driving distance on a single electric charge increased by approximately 7 km. Thus, it was expected that the driving performance of an EV with continuously variable transmission could be improved by downsizing the electric motor to reduce manufacturing costs. We will continue to conduct research on this matter in the future.

**Author Contributions:** Conceptualization, J.L.; methodology, J.L. and W.K.; project administration, J.L.; supervision, J.L.; validation, J.L.; investigation, W.K., D.K., D.L. and I.M.; data curation, W.K.; formal analysis, W.K.; writing—original draft preparation, W.K.; writing—review and editing, W.K., D.K., D.L., I.M. and J.L. All authors have read and agreed to the published version of the manuscript.

**Funding:** This research received no external funding.

**Institutional Review Board Statement:** Ethical review and approval are not applicable for this study not involving humans or animals.

**Informed Consent Statement:** Informed consent was obtained from all subjects involved in the study.

**Data Availability Statement:** The data presented in this study are available on request from the corresponding author.

**Acknowledgments:** This research was supported by Soongsil University, Republic of Korea.

**Conflicts of Interest:** The authors declare no conflict of interest.

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
