# Peer review of "Analysis of Optimal Shift Pattern Based on Continuously Variable Transmission of Electric Vehicle for Improving Driving Distance"

_applsci, doi:10.3390/app13021190_

Round 1
Reviewer 1 Report
· Change the word “Study” from the Title and replace as per the main theme of the article
· Highlight some advantages (numerical values if possible) through comparative analysis between ICE and EV in Introduction, and support such with relevant literature. And the same goes for disadvantages of EV also.
· Highlight more problems related to EV in Introduction section of the Paper.
· Justify through literature the high efficiency of EV as compared to the ICE, and define what wide operation range for EV
· Improve English Grammar and sentence structures of the Introduction section of the manuscript
· Rewrite the literature review portion of the Introduction section and highlight major issues related to the efficiency of motor.
· Add major issues related to the CVT, and justify them with operating point change, constant output area, and transmission response speed
· Figure 1 is not visible, improve its presentation
· Give equation numbers to the equations
· Write Specifications of EV used in this study Company B, and also write specifications for Argonne National Laboratory (ANL)
· Improve the presentation of the Results, justify the variations in the graphical results with valid references.
· Add more literature to support the main results of the manuscript.
· Highlight properly the main out-put results
· Make consistent presentation / appearance of Figures.
· Numerous grammatical errors and mistakes in sentence structures.
Author Response
Dear editor and reviewer
The authors would like to deeply thank the editor and reviewer for reviewing the paper and providing constructive remarks. We have revised the manuscript based on the reviewer’s and the editor’s comments. It has greatly helped us to improve our work.
Detailed answers to the review comments were provided in this document. The corresponding modifications in the manuscript are described at each answer, and also highlighted with blue color in the revised manuscript. Big modifications including corrections of English are also progressed in the revised manuscript.
So, please check the highlighted parts of the revised manuscript.
Sincerely yours,
Jin Wook Lee

Reviewer 2 Report
Review of the paper:
Study on Optimal Shift Pattern of Electric Vehicle with CVT for Improving the Driving Distance
Authors:
Wootaek Kim, Daekuk Kim, Dokyeong Lee, Iljoo Moon, and Jinwook Lee
The article presented for review is very interesting from an application point of view and because the research is mainly concerned with optimizing the performance of the electric vehicle's components, with the aim of extending the electric drive's operation and increasing its economy of use. The presented research complements the existing state of knowledge in the topic of electric vehicle operation, and to some extent, the results can provide a starting point for the creation of new mechatronic gearbox designs with higher application potential for use in new and existing electric vehicles.
Nevertheless, the paper submitted for review requires some clarifications and corrections to meet the conditions for a substantial scientific publication in the Applied Sciences Journal.
List of detailed comments below:
- The sections Title and Abstract: Please do not use abbreviations in these sections; there is a place for that in the text of the article.
- The section Introduction 1) I propose to cite specific scientific papers describing the stated advantages and disadvantages of electric vehicles; 2) when describing the advantages of CVT transmissions and mentioning other types of transmissions used in automobiles, it would be appropriate to supplement this section with a broader description of them, which would perhaps consequently provide a basis for selection of CVT in research; 3) I propose to describe more broadly the commercial electric vehicle model of Company B, possibly citing information on the Internet.
- Table 1: What does Cd Item mean? Please expand the description. Please also correct the tire size description.
- Section 3: The degree of convergence with experimental data should be used to gauge how reliable the developed model is. This then provides a realistic assessment of the outcomes and their applicability for additional synthesis in real-world projects.
- Figures 11 and 12—perhaps it is better to use the term Energy efficiency, instead of Fuel efficiency?
- The section Conclusion—I suggest rebuilding the content of this section, moving some of the content presented to the Results and Discussion section. Please avoid the following statements: ... confirmed good agreement between analysis and experiment data ... - what does good mean in practice? Please state specifically what this means.
Author Response

(The authors gave the same response as above.)

Reviewer 3 Report
I would like to thank the authors for their efforts in this work. Here some of the comments which I think should be considered:
1- In the introduction section, I found the literature review is weak and very brief. I would like to see a more critical review in related work. The critical review shall help other researchers to work in this subject. please expand your literature to be more critical. You might include a table summarizing the previous papers outcomes
2- In section 2.3.1 the CVT applied EV modeling, this section does not explain the model very well. It should be rewritten in a way to explain in details your applied modeling approach.
3- Kindly include a mathematical model explaining your applied modeling in the previous comment. State clearly your assumptions to build your model
4- in section 3, please explain how did you make this comparison and based on what
5- I would recommend to add one more section benchmarking the performance of your applied approach with other proposed by researchers. highlighting the key features of the proposed approach.
6- In the conclusion, provide what are the possible future research in this subject
Author Response

(The authors gave the same response as above.)
